# Marine Algae-Derived Porous Carbons as Robust Electrocatalysts for ORR

**Yang Li [1], Xianhua Liu [1,*], Jiao Wang [1], Li Yang [1], Xiaochen Chen [2,*], Xin Wang [3] and Pingping Zhang [4]**

[1] Tianjin Key Laboratory of Indoor Air Environmental Quality Control, School of Environmental Science and Engineering, Tianjin University, Tianjin 300354, China
[2] Fujian Provincial Engineering Research Center of Rural Waste Recycling Technology, College of Environment and Resources, Fuzhou University, Fuzhou 350108, China
[3] Department of Microbiology, Miami University, Oxford, OH 45056, USA
[4] College of Food Science and Engineering, Tianjin Agricultural University, Tianjin 300384, China
\* Correspondence: lxh@tju.edu.cn (X.L.); chenxiaochen@fzu.edu.cn (X.C.)

**Abstract:** Large quantities of marine algae are annually produced, and have been disposed or burned as solid waste. In this work, porous carbons were prepared from three kinds of marine algae (*Enteromorpha*, *Laminaria*, and *Chlorella*) by a two-step activation process. The as-prepared carbon materials were doped with cobalt (Co) and applied as catalysts for oxygen reduction reaction (ORR). Our results demonstrated that Co-doped porous carbon prepared from *Enteromorpha* sp. (denoted by Co-PKEC) displayed excellent catalytic performance for ORR. Co-PKEC obtained a half-wave potential of 0.810 V (vs. RHE) and a maximum current density of 4.41 mA/cm$^2$, which was comparable to the commercial 10% Pt/C catalyst (E$_{1/2}$ = 0.815 V, J$_d$ = 4.40 mA/cm$^2$). In addition, Co-PKEC had excellent long-term stability and methanol resistance. The catalytic ability of Co-PKEC was evaluated in a one-chamber glucose fuel cell. The maximum power density of the fuel cell equipped with the Co-PKEC cathode was 33.53 W/m$^2$ under ambient conditions, which was higher than that of the fuel cell with a 10% Pt/C cathode. This study not only demonstrated an easy-to-implement approach to prepare robust electrochemical catalyst from marine algal biomass, but also provided an innovative strategy for simultaneous waste remediation and value-added material production.

**Keywords:** marine algae; biomass; oxygen reduction reaction; catalyst; fuel cell

## 1. Introduction

Marine algae are annually produced in large quantities, and they are typically burned or landfilled as biowastes. For example, *Enteromorpha prolifera* (*E. Prolifera*) green tide has become one of global ecological and environmental problems [1]. The world's largest scale of *E. Prolifera* green tide broke out in the summer of 2008 in China's Yellow Sea consisted of more than 100 million tons of drifting algal biomass. Two methods have been employed to addressing the environmental issue of green tide: mitigation measures and preemptive management [2]. However, mitigation measures may have potential negative impacts to the local ecosystem, and preemptive management is difficult to implement with increasing agricultural activities. Therefore, alternative approaches are needed to addressing the issue of massive algal blooms. It would benefits economy, environment and our society if technologies are available to convert the large amount of algal biomass into functional high-value products [3].

Fuel cells have been considered as one of the promising technology for energy conversion since it can directly converts chemical energy into electrical energy without combustion [4–6]. At present,

research on fuel cells has made great progress, but its short life, low efficiency and high cost limit its large-scale commercial application [7,8]. Oxygen reduction reaction (ORR) is one of the most important chemical reactions in electrochemical devices, which plays a critical role for energy transformation efficiency in fuel cells. However, the high ORR overpotential resulting from sluggish kinetics requires large amount of precious metal catalyst, such as Pt, to promote the reaction activity and durability. Therefore, the development of low cost, high activity and high stability non-precious metal catalysts for ORR is of great significance for pushing forward the large-scale commercial application of fuel cells [4,9,10]. Among the alternative catalysts studied so far, carbon-based materials have grabbed much attention owing to the balance between cost, activity, and durability. Carbon-based materials can be produced from various precursors, and biomass is one of abundant, renewable and green sources of carbon material. Biomass-derived carbon materials have been widely applied in nearly all kinds of energy conversion and storage applications. Researchers have evaluated catalytic performances of non-metal (N, P, S, B) and transition metals (Co, Fe, Ni, Mn, etc.) doped graphene [11,12], carbon nanotubes [13,14] and multi-walled carbon nanotubes [15,16].

Development of carbon-based catalysts from marine algae biomass could simultaneously ease the burden of ocean pollution and solve the challenges of the high cost, and limited availability of precious metals. Algae such as Laver, Sargassum, etc., has been applied as a precursor for the preparation of heteroatom-doped carbon material because of its abundant protein and strong ion exchange capacity [17,18]. Monoatomic Fe electrocatalyst was prepared by using Laver as the precursor [19], and exhibited excellent catalytic activity and high stability compared with Pt/C catalyst. Alginate biomass sodium alginate was used as source material to prepare a series of defective carbon catalysts, and demonstrated high ORR activity and good electrical conductivity [20]. These previous studies demonstrate that marine algae represents a potential precursor for high performance carbon-based materials. However, very limited studies are available in the literature concerning the activation treatments to enhance their catalytic ability. In addition, effect of marine algae species on the catalytic performance has not been systematically investigated.

The primary objective of this investigation is to perform activation process on different marine algae biomasses in order to find a low-cost way to turn waste into high-value products. This study will provide an easy-to-implement approach to prepare robust ORR catalyst from renewable marine algal biomass, and provides an innovative strategy for simultaneous wastes remediation and production of value-added products.

## 2. Results and Discussion

### 2.1. ORR Performance of PKEC, PKCC and PKKC

Figure 1A shows the CV curves of PKEC, PKCC and PKKC. For all the three algal biomass-derived catalysts, an obvious reduction peak was observed at 0.7 V (vs. RHE) under saturated oxygen conditions. This peak disappeared when the electrolyte solution was purged with $N_2$, exhibiting that this peak corresponded to the reduction of $O_2$. Among all the three catalysts, PKEC displayed the lowest over-potential for ORR and the largest ORR reduction current density, which indicated that the ORR catalytic activity increased in the order of PKKC < PKCC < PKEC. Figure 1B displays the polarization current densities of the three algal biomass-derived catalysts. At the potential of 0.4 V (vs. RHE), the current densities were in the order of PKKC ($-2.6$ mA/cm$^2$) < PKCC ($-3.2$ mA/cm$^2$) < PKEC ($-3.9$ mA/cm$^2$), which is consistent with CV measurements.

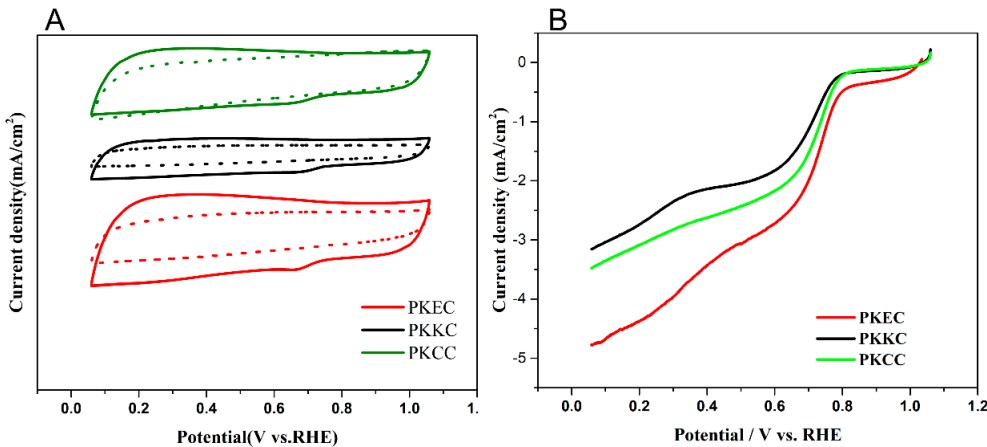

**Figure 1.** Electrochemical measurements of algal biomass derived catalysts on an RDE. (**A**) CV curves of PKEC, PKCC and PKKC in $O_2$-saturated (solid lines) or $N_2$-saturated (dotted lines) 0.1 M KOH aqueous solutions at a scan rate of 100 mV/s. (**B**) LSV curves of above three catalysts in $O_2$ saturated 0.1 M KOH at a sweep rate of 10 mV/s and rotation speed of 1600 rpm. Catalyst loading: 0.2 mg/cm$^2$.

Figure 2 shows LSV curves of different catalysts at various rotation speeds of the electrode. The cathodic current increased with the increase of rotating speed, and $j^{-1}$ and $\omega^{-1/2}$ are in a positive linear relationship. The obtained values of n for PKCC and PKKC calculated from the K-L plots were 1.47–2.95 in the range of 0.2–0.6 V, indicating that the oxygen reduction process was mainly based on a two-electron transferring reaction. The number of electron transfer of the PKEC catalyst was 2.68–4.10 in the range of 0.2 V to 0.6 V, demonstrating the oxygen reduction on PKEC was occurred mainly through a more efficient four-electron transferring process.

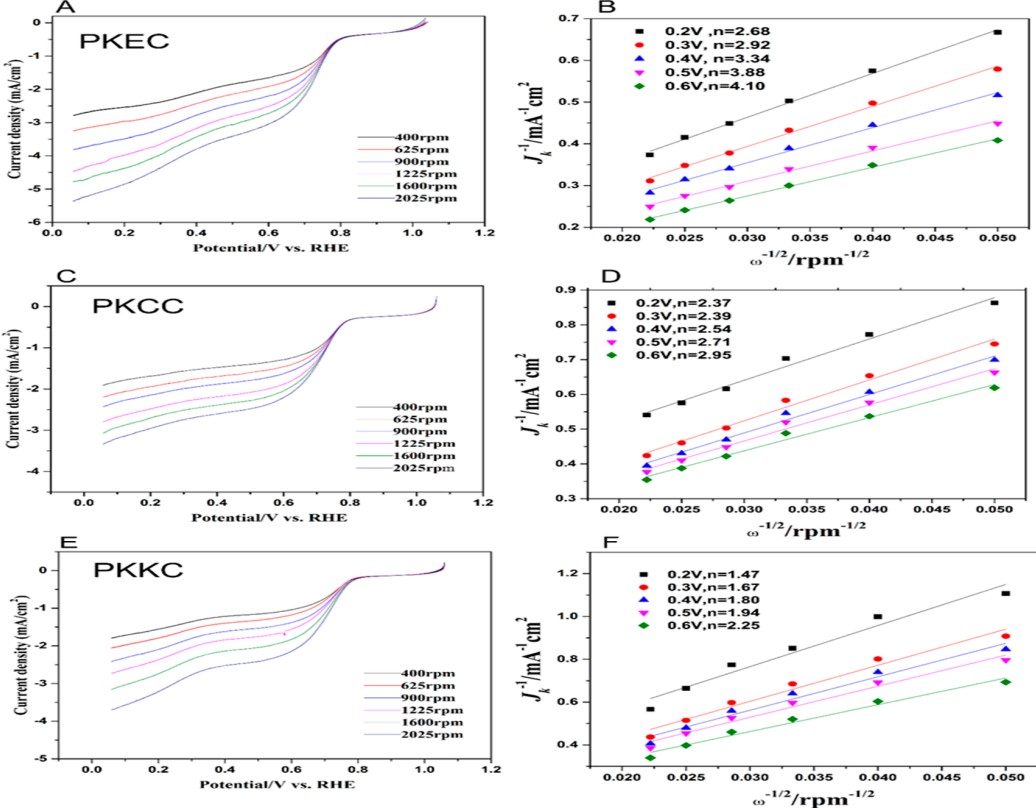

**Figure 2.** LSV curves of PKEC (**A**), PKCC (**C**) and PKKC (**E**) in $O_2$ saturated 0.1 M KOH at different rotation speeds and K-L diagrams of PKEC (**B**), PKCC (**D**) and PKKC (**F**) at different potentials.

## 2.2. ORR Performance of Co-PKEC

### 2.2.1. CV and LSV

Figure 3A shows the CV curves of commercial Pt/C (10 wt%), PKEC and Co-PKEC. All the three catalysts exhibited obvious ORR peaks in the $O_2$-saturated 0.1 M KOH. However, the ORR peak of Co-PKEC was more pronounced than that of the PKEC and commercial Pt/C. Figure 3B shows the LSV curves of Co-PKEC, PKEC and commercial Pt/C (10 wt%) at a rotation speed of 1600 rpm in 0.1 M KOH. Co-PKEC displayed a larger diffusion limit current density ($J_d$ = 4.41 mA/cm$^2$) and an earlier half-wave potential ($E_{1/2}$ = 0.810 V vs. RHE) than those of commercial Pt /C ($J_d$ = 4.40 mA/cm$^2$, $E_{1/2}$ = 0.815 V vs. RHE), confirming that doping PKEC with cobalt remarkably enhanced ORR performance.

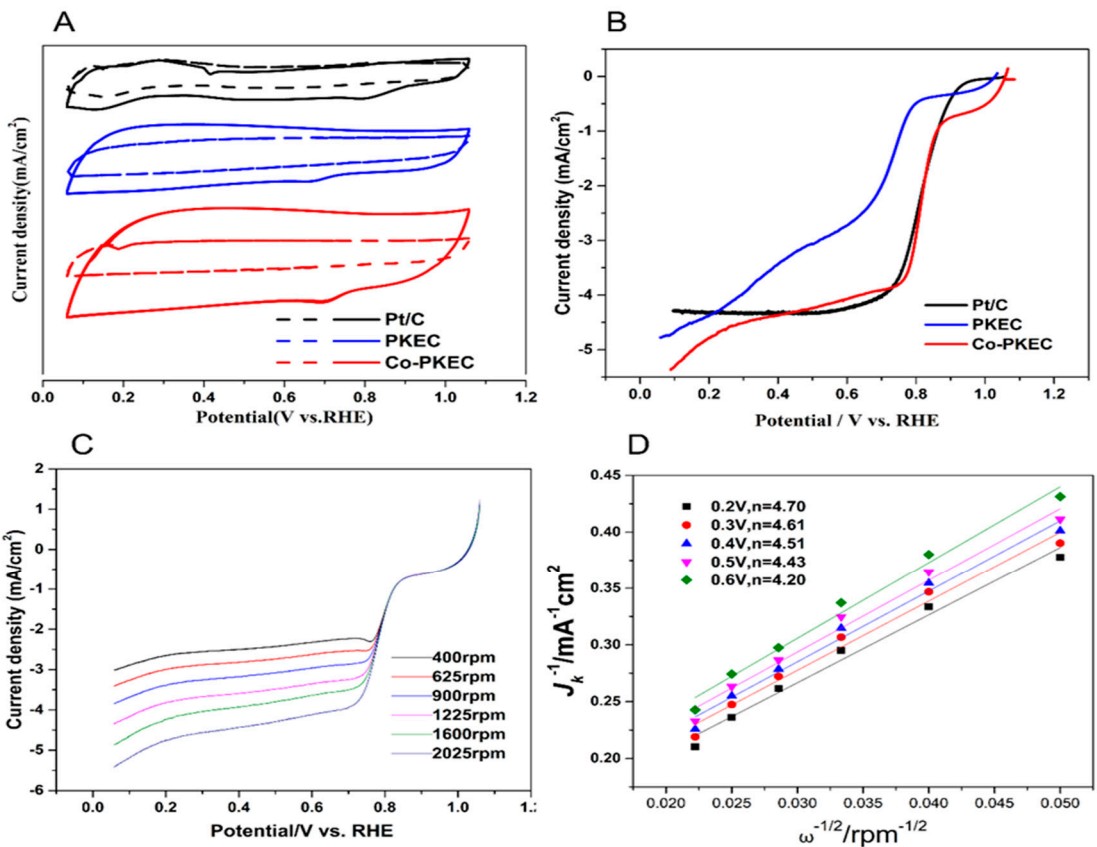

**Figure 3.** Electrochemical measurements of Co-PKEC on an RDE. (**A**) CV curves of Co-PKEC in $O_2$-saturated (solid lines) or $N_2$-saturated (dotted lines) 0.1 M KOH aqueous solutions at a scan rate of 100 mV/s. (**B**) LSV curves of Co-PKEC, PKEC and Pt/C in $O_2$ saturated 0.1 M KOH at a sweep rate of 10 mV/s and rotation speed of 1600 rpm. (**C**) LSV curves of Co-PKEC at various rotation speeds of electrode. (**D**) K-L diagrams at different potentials. Ca

The ORR kinetic mechanism of Co-PKEC was further studied by RDE voltammetry method. Figure 3C shows that the current density of Co-PKEC increases with the increase of RDE rotation speed, and there is a good linear relationship between $J_k^{-1}$ and $\omega^{-1/2}$. The electron transfer number (n) of Co-PKEC in alkaline medium was 4.2–4.7, demonstrating that the ORR on the surface of Co-PKEC electrode was mainly carried out by a near four-electron process (Figure 3D). Theoretically the electron transfer number (n) for ORR is no more than 4. Our experimental results were slightly greater than 4 maybe due to the fact that the Koutecky–Levich (K-L) method used here to determine n was developed under certain assumptions and conditions. According to others' previous studies performed for a wide variety of electrocatalysts, the n values obtained by the K-L method sometimes exceed theoretical limits [21,22]. Table 1 lists the parameters of reported ORR catalysts derived from biomass and doped

with various heteroatoms. Co-PKEC exhibits high BET specific surface area, pore volume and high ORR activity.

**Table 1.** ORR catalytic performance of biomass derived carbons doped with various heteroatoms under alkaline conditions.

| Biomass | Doped Elements | Activator | Specific Surface Area $(m^2/g)$ | Pore Volume $(cm^3/g)$ | $E_{Onset}$ (V vs. RHE) | $E_{1/2}$ (V vs. RHE) | $J_d$ $(mA/cm^2)$ | Electron Transfer Number (n) | Article |
|---|---|---|---|---|---|---|---|---|---|
| Enteromorpha | Co,N,P | $H_3PO_4$ KOH | 1783.83 | 1.57 | 1.05 | 0.81 | 4.4 | 4.5 | This study |
| Coconut shell | N,P | $H_3PO_4$ | 1071 | 0.9 | 0.95 | 0.76 | 5.3 | 3.7 | [23] |
| Eggplant | N | $NH_3$ | 1969 | - | 0.935 | 0.636 | 5.5 | 3.87 | [24] |
| Soybean | N,S | $ZnCl_2$ | 949 | 0.42 | 0.95 | 0.84 | 5 | 4 | [25] |
| Pomelo peel | Co,N | KOH | 2091 | 1.28 | 0.88 | 0.97 | 5.2 | 3.9 | [26] |
| Pulsatilla chinensis | P | - | - | - | 0.94 | 0.773 | 4.36 | 3.6-4.6 | [27] |
| Laver | Fe | KOH | 1070 | 0.95 | - | 0.87 | 5 | 4 | [28] |
| Corn silk | Fe,N | $NH_3$ | 592 | 0.54 | 0.957 | 0.927 | 5 | 3.87-4 | [29] |
| Bean dregs | Fe | - | 517 | 0.43 | 0.98 | 0.87 | 4 | 3.5 | [30] |
| Pork liver | Fe,N | - | 700 | - | 0.985 | 0.82 | 5 | 3.7 | [31] |
| Porcine blood | N | - | - | - | 0.915 | 0.81 | 3.4 | 3.7 | [32] |
| Chicken feather | N,S,Zn | $NH_3$ | 829 | - | 1.0 | 0.94 | 3.4 | 3.98 | [33] |

### 2.2.2. Stability Test and Methanol Poisoning Resistance Test

The stability of catalysts in fuel cells has attracted much attention. The electrochemical stabilities of Co-PKEC and commercial Pt/C (10% Pt) in alkaline medium were compared by amperometric chronometry. As shown in Figure 4A, the chronoamperometric curves in 0.1 M oxygen saturated KOH solution clearly reflect that Co-PKEC has a slower decay rate and better stability than commercial Pt/C. After 25,000 s test, the current of Co-PKEC can still reach 82.7% of the initial value, which is much higher than that of commercial 10% Pt/C (72.0% of the initial value). This indicated that in alkaline medium, Co-PKEC has better electrochemical stability than commercial Pt/C. The reason why the activity of commercial Pt/C is easy to decrease is that platinum nanoparticles are easy to coagulate and dissolve.

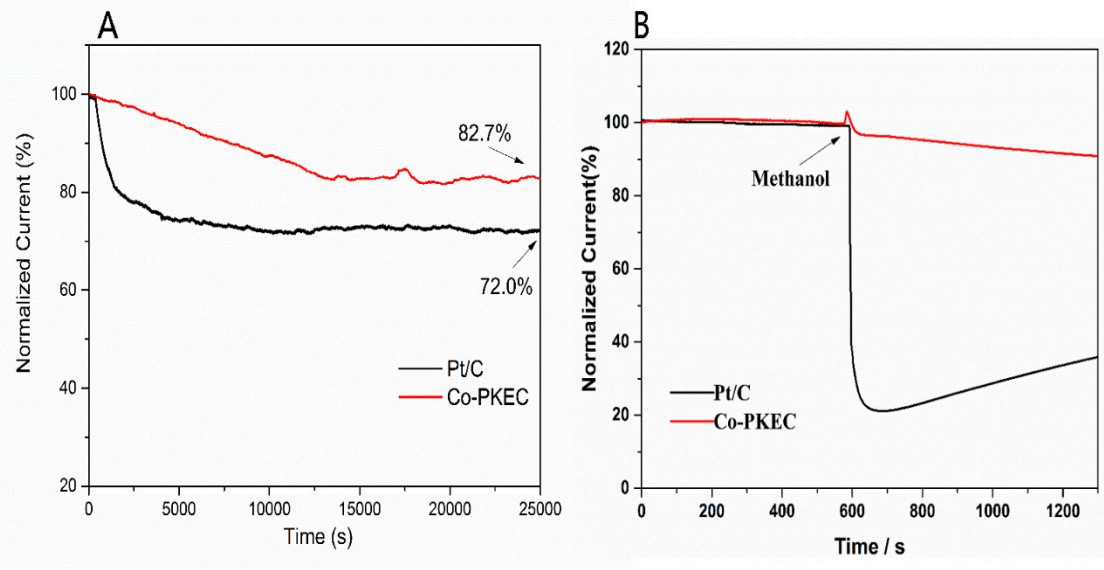

**Figure 4.** Stability test (**A**) and methanol poisoning resistance test (**B**) of Co-PKEC.

Methanol poisoning resistance test was used to test the substrate selectivity of Co-PKEC. Cathode catalyst is expected to selectively catalyze oxygen reduction reaction and is resistant to methanol interference. Figure 4B is I-t curves of Co-PKEC and commercialized Pt/C electrodes in 0.1 M KOH solution saturated with oxygen (disc speed 800 rpm). Chronoamperometry was used to detect the

possible cross-effect of methanol on the catalysts by adding 3 M methanol solution at 600 s. Figure 4B shows that the oxygen reduction current of commercial Pt/C decreased sharply when methanol was added, but no obvious change was observed at the Co-PKEC electrode at the same conditions. These results exhibit that Co-PKEC catalyst has super methanol resistance ability than commercial Pt/C catalyst in alkaline medium.

## 2.3. Material Characterization

### 2.3.1. SEM Characterization and Pore Structure Analysis

Figure 5 shows the SEM images of Co-PKEC with different magnification. It can be seen that Co-PKEC contains porous surfaces with empty spaces or pores that can allow external matter to penetrate.

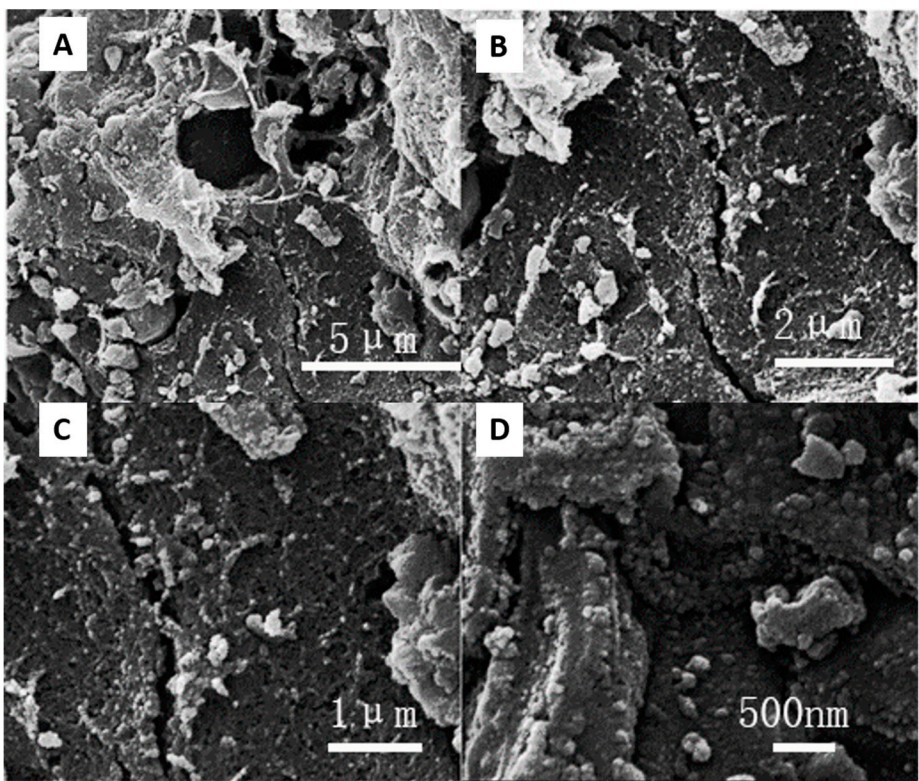

**Figure 5.** SEM of Co-PKEC with different magnification.

To further understand the pore structure, $N_2$ adsorption and desorption isotherms and pore size distribution of catalysts PKEC and Co-PKEC are shown in Figure 6. The variation of specific surface area and pore structure of samples before and after Co was analyzed by $N_2$ adsorption-desorption isotherm and pore size distribution curve. The $N_2$ adsorption-desorption isotherms of both PKEC and Co-PKEC are close to type IV, and the $H_4$ hysteresis loops in the isotherms at P/P = 0.4 indicated that the catalysts are microporous and mesoporous materials, and many interconnected mesoporous structures and narrow crack pore structures. This data is consistent with the SEM results of the material. The BET specific surface areas of Co-PKEC and PKEC are 1783.83 $m^2$/g and 2709.84 $m^2$/g, respectively. The pore size and pore volume of Co-PKEC are smaller, which may be due to the embedding of metal nanoparticles between the carbon nanosheets, resulting in the decrease of BET specific surface area. Table 2 shows that the pore volume of the PKEC Catalyst doped with Co decreases from 2.49 $cm^3$/g to 1.57 $cm^3$/g. Pore size distribution curves demonstrates that the pore size structures are

mainly mesoporous. The average pore size of Co-PKEC and PKEC are 3.52 and 3.67 nm, respectively. The change of pore size structure is supposedly due to the addition of metal nanoparticles.

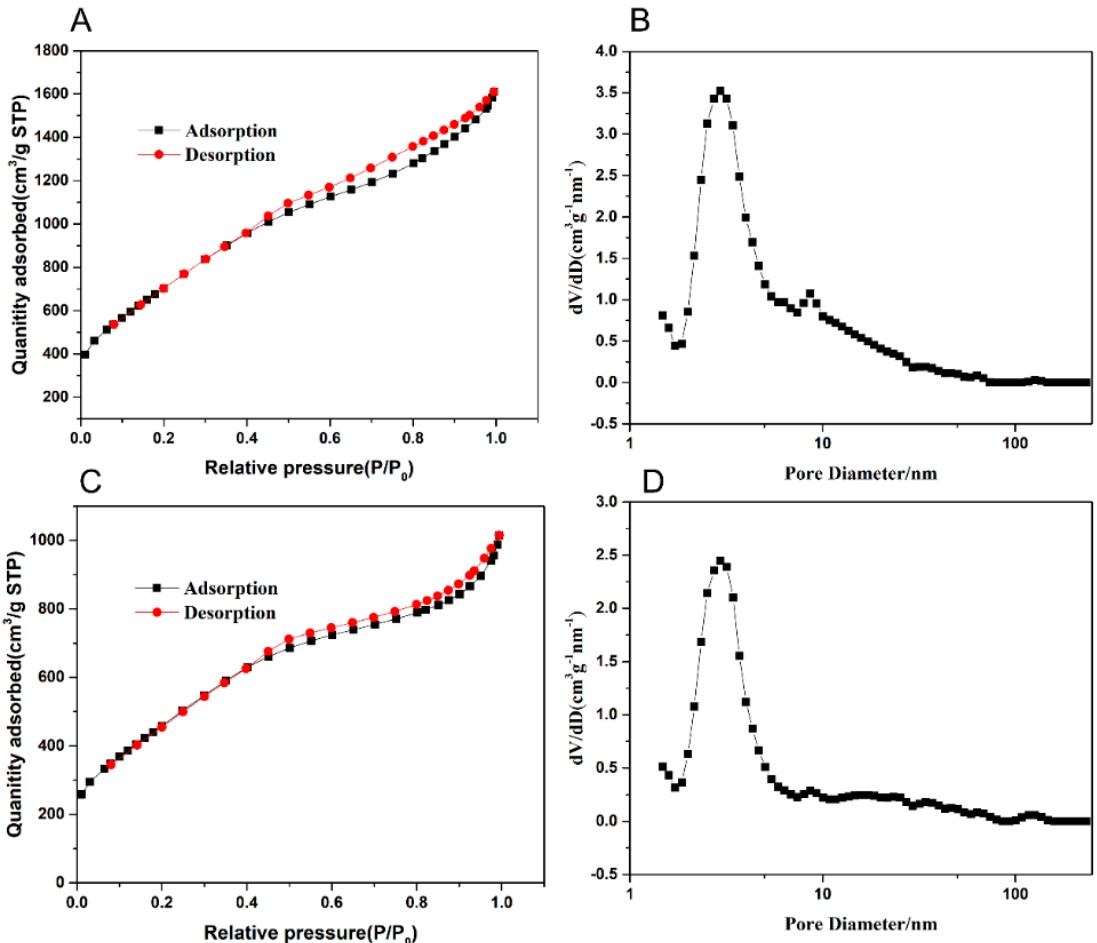

**Figure 6.** N$_2$ adsorption desorption isotherm and pore size distribution of PKEC (**A**,**B**) and Co-PKEC (**C**,**D**).

**Table 2.** Surface chemical composition of PKEC and Co-PKEC.

| Sample | Carbon (At. %) | Oxygen (At. %) | N Total (At. %) | P Total (At. %) | Co (At. %) |
|---|---|---|---|---|---|
| PKEC | 88.34 | 9.72 | 1.6 | 0.34 | — |
| Co-PKEC | 88.16 | 9.51 | 1.45 | 0.38 | 0.5 |

### 2.3.2. Structural Characterization of Materials

XPS (X-ray photoelectron spectroscopy) analyses of PKEC and Co-PKEC were carried out to determine the composition and surface chemical composition of the materials. Figure 7 shows there were obvious C 1s characteristic peaks at 286 eV, O 1s characteristic peaks at 540 eV, P 2p characteristic peaks at 130 eV and N 1s characteristic peaks at 400 eV. Compared with the PKEC catalyst, the peak of Co-PKEC catalyst at about 800eV belongs to Co 2 p. The contents of C, O, N and P in PKEC catalysts are 88.34%, 9.27%, 1.6% and 0.34% according to the Table 3. The contents of C, O, N and P in Co-PKEC catalysts are 88.16%, 9.51%, 1.45% and 0.38% respectively. It can be seen that the addition of Co has not changed significantly. The addition process of metal has a significant effect on the element content of carbon materials, which indicates that N and P elements have been successfully incorporated into the framework of carbon structure.

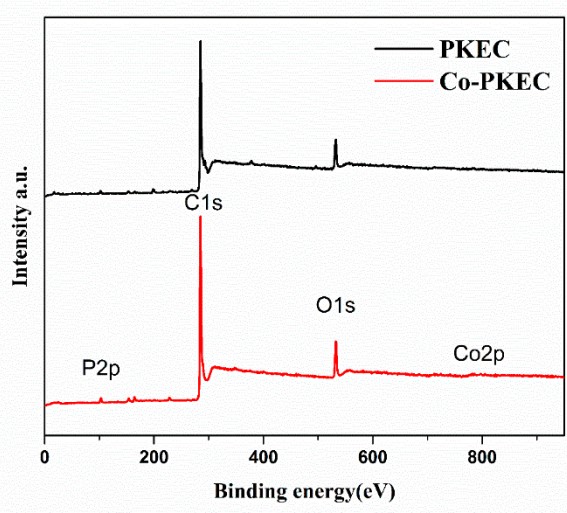

**Figure 7.** XPS energy spectrum of PKEC and Co-PKEC.

**Table 3.** Composition of nitrogen elements in PKEC and Co-PKEC.

| At. % | Pyridinc-N | Pyrrolic-N | Graphitic-N | Pyridinc-N-oxide |
|---|---|---|---|---|
| PKEC | 10.89 | 63.15 | 6.23 | 19.74 |
| Co-PKEC | 16.54 | 43.44 | 10.35 | 29.67 |

Figure 8 shows the XPS spectra of N 1s and P 2p of PKEC and Co-PKEC. N 1s spectra of the samples (Figure 8A,C) indicate that there are mainly four forms of nitrogen elements in the samples: pyridine-N, pyrrole-N, graphite-N and pyridine-N-oxide [34]. In the preparation process, there were no other nitrogen source was introduced into the samples besides of CoPc. Therefore, the nitrogen element in the samples may came from the pyrolysis of CoPc and the algae biomass itself. Table 3 lists the total content of pyridine nitrogen and graphite nitrogen in Co-PKEC (26.89%), which is significantly higher than that in PKEC (17.12%). These N atoms supposedly act as active sites for ORR catalysis. Figure 8C indicates that the N-peak position of Co-PKEC catalyst shifts to a higher binding energy under cobalt loading. Based on its possible coordination, this may be the result of electron transfer between transition metals and nitrogen groups. This implies that the conversion of pyridine-N-oxide can lead to the increase of pyridine-N and graphite-N contents with the addition of Co. That is to say, the conversion of inert N species to active N species increases the catalytic sites in the sample. Figure 8B,D are the peaks of P 2p. The existing forms of phosphorus element can be analyzed from the peaks of P-C, P-N and P-O. Comparing the peaks before and after doping, it can be seen that the doping of Co has an effect on the content and morphology of P.

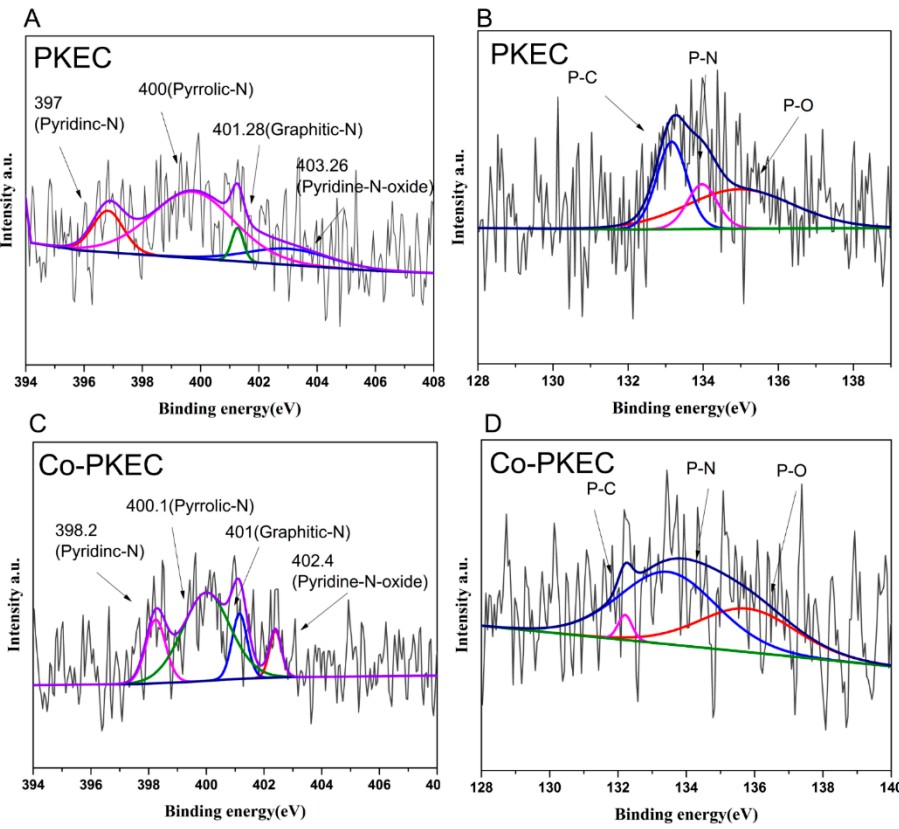

**Figure 8.** N 1 s (**A**,**C**) and P 2p (**B**,**D**) XPS spectra of PKEC and Co-PKEC.

Figure 9 shows the Co 2p spectrum of Co-PKEC, and the fitting peaks appear at 781.83 eV, 785.6 eV, 789.04 eV, 796.71 eV, 801.35 eV and 804.79 eV. The weak bond energies of 785.6 eV and 801.35 eV correspond to Co 2p$^{3/2}$ and Co 2p$^{1/2}$, respectively, indicating that a small amount of CO exists in the catalyst, while the diffraction peaks of 796.7 eV (Co 2p $^{1/2}$) and 781.83 eV (Co 2p $^{3/2}$) in the Co 2p XPS spectra correspond to Co$_3$O$_4$. These results demonstrate that Co element exists mainly in the form of Co and Co$_3$O$_4$.

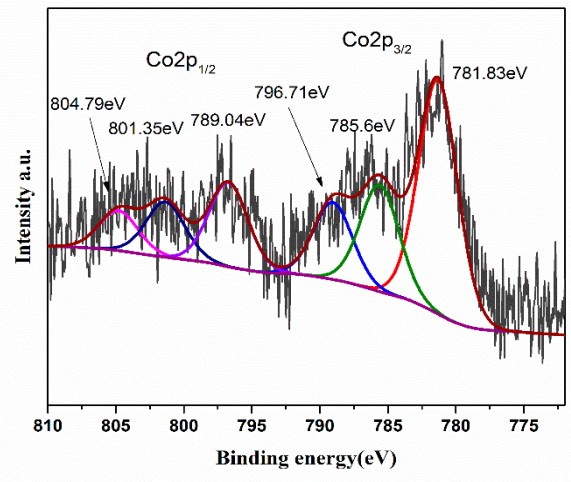

**Figure 9.** Co 2p peak curve of XPS spectrum of Co-PKEC.

### 2.4. Performance of Co-PKEC in Fuel Cell

Figure 10A displays the power density curves of an alkaline fuel cell equipped with three different cathodes, Co-PKEC, Pt/C, and commercial activated carbon (AC), respectively. When using 3 M KOH and 1 M glucose as the substrate, the maximum power densities of the fuel cell with different cathodes were as follows: commercial AC ($17.59 \pm 2.27$ W/m$^2$) < Pt/C ($26.83 \pm 2.29$ W/m$^2$) < Co-PKEC ($33.53 \pm 1.35$ W/m$^2$). The current densities were in the same order: commercial AC (68.56 A/m$^2$) < Pt/C (87.02 A/m$^2$) < Co-PKEC (96.53 A/m$^2$). The maximum power density of the catalyst Co-PKEC was much higher than that of the AC air cathode fuel cell, and even better than those previously reported direct glucose alkaline fuel cells. In addition, the open circuit voltage (0.8 V) of the fuel cell equipped with Co-PKEC as cathode was also higher than those of the fuel cell equipped with the Pt/C cathode (0.69 V) or the commercial AC cathode (0.66 V). Polarization curves of the anode and cathode (Figure 10B) shows that the anodic polarization curves of the different fuel cells are almost completely coincident, while the cathode polar polarization curves are significantly different, indicating that the increased power density output comes from improved cathode performance. At the same time, it can be seen that the cathode open circuit potential of the catalyst Co-PKEC is significantly higher than other catalysts, which further confirms that the Co-PKEC catalyst has high catalytic oxygen reduction activity. It can be used as an ideal environmentally friendly catalyst for replacing noble metal catalysts in alkaline fuel cells.

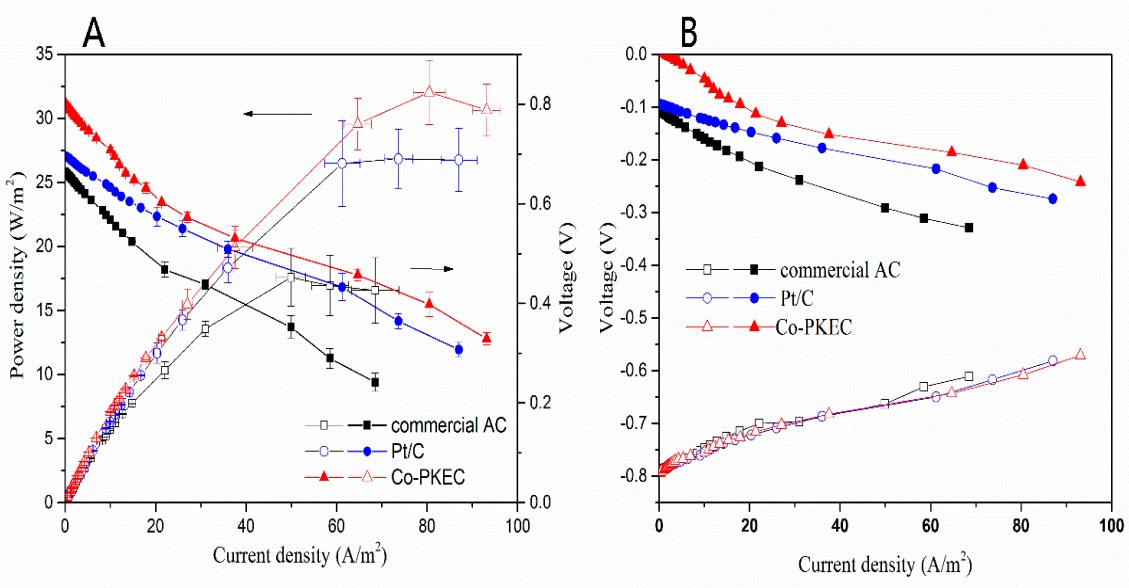

**Figure 10.** Power density curve (**A**) and electrode polarization curves (**B**) of fuel cell with Co-PKEC, activated carbon and commercial Pt/C as cathode electrodes.

## 3. Materials and Methods

### 3.1. Chemicals

Marine algae powders of enteromorpha (*Enteromorpha* sp.), kelp (*Laminaria* sp.), and chlorella (*Chlorella* sp.) were obtained from an online store (Yi Chongtang, Qingdao, P.R. China). Nafion solution (5 wt%) and commercial 10% Pt/C catalyst were purchased from DuPont Corporation (Midland, MI, USA) and Johnson Matthey Company (China). H$_3$PO$_4$, KOH, anhydrous alcohol, cobalt phthalocyanine (CoPc) and glucose were all analytical reagents and used without further purification.

### 3.2. Catalyst Preparation

In this study, three types of algal biomasses (Enteromorpha, Kelp and Chlorella) were selected for catalyst preparation. And $H_3PO_4$-KOH co-activation, a method for the activation of algae biomass was selected [35,36]. The specific steps were as follows:

$H_3PO_4$ activation: 10 g of three types of algae powders were weighed separately, and added into 20 g of 50% wt $H_3PO_4$ solution, then placed in a drying oven at 80 °C for 12 h. The raw material was placed in a muffle furnace and activated at a temperature of 450 °C for 90 min. After the reaction, the black carbon material was taken out and repeatedly washed with distilled water until the wash water was neutral. Finally, it was transferred to a vacuum drying oven and dried at 80 °C for 1 h.

KOH activation: The black carbon material obtained in the previous step was mixed with KOH in a mass ratio of 1:4. After full grinding and stirring, it was placed in a corundum boat and placed in a tubular furnace under nitrogen atmosphere. The temperature raised to 800 °C at a rate of 4 °C/min. After 1 h at 800 °C, it was naturally cooled to room temperature under nitrogen atmosphere. The carbon material was washed several times with deionized water. Next, a certain amount of 1 M $HNO_3$ solution was added and stirred for 4 h, then it was washed repeatedly with deionized water and ethanol until it was neutral. Finally, after drying for 12 h in a vacuum drying chamber at 80 °C, the $H_3PO_4$-KOH co-activated enteromorph, chlorella and kelp carbons were obtained, which were named PKEC, PKKC and PKCC, respectively.

Preparation of Co-doped porous carbons: Co-doped porous carbons were synthesized by calcination at high temperature using various algal biomass carbons (PKEC, PKKC and PKCC) as carbon support and cobalt phthalocyanine (CoPc) as cobalt source [37,38]. The specific steps were as follows: 50 mL of 0.1 M KOH was weighed and heated to 60 °C on a magnetic heating stirrer. Then, 0.1 g sample of algal biomass carbon was added under continuous stirring. After 30 min, 10 mL of 3 g/L CoPc was dripped into the above carbon-containing solution at a uniform speed and stirred continuously for 4 h. The mixture was centrifuged to collect the solid. The supernatant was colorless, indicating the complete adsorption of CoPc. The collected solid was dried in a vacuum oven at 80 °C for 12 h. After further thermal treatment at 800 °C for 2 h under a nitrogen atmosphere, it was naturally cooled to room temperature to obtain Co-doped porous carbon (Co-PKEC, Co-PKKC and Co-PKCC). Figure 11 shows the schematic procedures for various algal biomass carbons doped with cobalt.

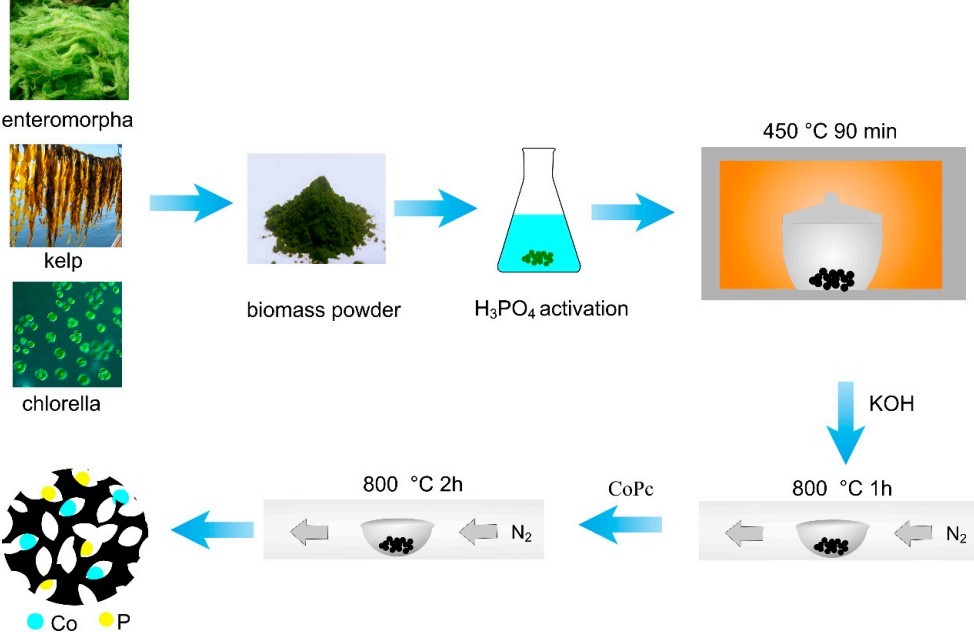

**Figure 11.** Schematic presentation of the synthesis procedure for three algal biomass derived carbons doped with cobalt.

### 3.3. Electrochemical and Physical Measurements

Linear sweep voltammetry (LSV), Cyclic voltammetry (CV) and chronoamperometric curves were measured at room temperature by a CHI 660 E three-electrode workstation (Shanghai, China). The working electrode was a glass carbon (GC) electrode with a diameter of 4 mm. Platinum wire and Hg/HgO (1M KOH) electrode were used as counter electrode and reference electrode, respectively. The catalyst ink was prepared by adding 5 mg catalyst and 50 μL Nafion solution (5 wt%) in 8 mL 1:1 (*v/v*) deionized water/ethanol and. After 30 min of ultrasonic treatment, a uniform catalyst ink was obtained. Then, 5 μL well-dispersed catalyst ink was deposited on the polished GC electrode and the loaded catalysts was 0.25 mg/cm$^2$. The potential of Hg/Hgo electrode is finally converted into reversible hydrogen electrode (RHE) [39]. LSV and CV were measured in 0.1 M KOH solution saturated with $O_2$ or $N_2$. The chronoamperometric measurements were carried out for 10 h at constant potential in 0.1 M KOH solution saturated with $O_2$. The rotational speed of the disc was 800 rpm. 3 M methanol was added to 0.1 M KOH solution saturated with $O_2$, and methanol crossover was measured after 500 s.

The number of electrons transferred (n) can be calculated from Koutecky-Levich (K-L) equation by rotating the electrode at different rates [40]:

$$J^{-1} = J_K{}^{-1} + J_L{}^{-1} = B^{-1}\omega^{-1/2} + J_K{}^{-1} \tag{1}$$

$$B = 0.2nFD_0{}^{2/3}C_0 v^{-1/6} \tag{2}$$

Among them, $J$, $J_K$ and $J_L$ are the measured current, the kinetic and diffusion limited current densities, respectively; $\omega$ represents the electrode rotation rate (r/min), F is the Faraday constant (96,485 C/mol); $D_0$ represents the diffusion coefficient of $O^2$ in 0.1 M KOH ($D_0 = 1.9 \times 10^{-5}$ cm$^2$/s); $v$ represents the kinematics viscosity of KOH ($v = 0.01$ cm$^2$/s) and $C_0$ is the concentration of $O^2$ in the electrolyte ($C_0 = 1.2 \times 10^{-6}$ mol/cm$^3$).

The size and shape of the samples were analyzed using a scanning electron microscopy (SEM, HITACHI/S-4800). The specific surface areas were measured from the nitrogen adsorption/desorption using the Brunauer-Ennett-Teller (BET) method (ASAP 2020 gas adsorption apparatus, Micromeritics, Norcross, GA, USA.). X-ray photoelectron spectroscopy and X-ray diffraction analyses were performed by an X-ray photoelectron spectrometer (Escalab 250Xi, Thermo Scientific, Waltham, MA, USA) and a D8 Advance X-ray diffractomer (Bruker, Berlin, Germany), respectively.

### 3.4. Fuel Cell Assembly and Performance Characterization

A one-chamber alkaline fuel cell (12 mL) with a MV/AC/Ni anode and a three-layer air cathode was constructed as previously reported [1,5,6,41,42]. Co-PKEC and AC air cathodes were fabricated with Co-PKEC and AC as catalysts, respectively. The electrolyte was 1 M glucose and 3 M KOH solution. To determine the fuel cell electrochemical performance, polarization curve and power density measurements were conducted by varying the corresponding external resistance from 9000 Ω to 10 Ω.

## 4. Conclusions

In this study, we demonstrate the feasibility of converting highly available algal waste into a series of ORR electrocatalysts. The effect of the different algal biomass on the ORR catalytic performance was investigated, and the obtained samples were thoroughly characterized. Among all the products, Co-PKEC has the best electrochemical performance. It not only has a highly porous structure, but also has excellent ORR activity, methanol resistance and catalytic stability. It also demonstrates an ideal application prospect in low temperature fuel cells and can be considered as a promising alternative to commercial precious metal catalysts such as Pt/C. In addition, our experiments showed that the $H_3PO_4$/KOH co-activation method can be employed to prepare porous carbons with excellent properties

from algal biomass. This study provided a simple and easy to implement route to convert the algal wastes into valuable products, which benefits the sustainable development of our society.

**Author Contributions:** Conceptualization, Y.L. and X.L.; methodology, J.W., X.C., X.W., L.Y., Y.L.; formal analysis, X.C., P.Z., L.Y., Y.L.; data curation, J.W., X.C., X.W., Y.L.; writing—original draft preparation, Y.L., X.L.; writing—review and editing, P.Z., L.Y.; supervision, X.L.

**Funding:** This work was partially supported by the National Key R&D Program of China (Grant #2017YFC1404500), and Tianjin University-Fuzhou University Innovation Cooperation Project (No. 2019XZC-0040).

**Acknowledgments:** The authors would like to thank all the laboratory technicians.

**Conflicts of Interest:** There are no conflicts to declare.

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
