# Peer review of "Marine Algae-Derived Porous Carbons as Robust Electrocatalysts for ORR"

_catalysts, doi:10.3390/catal9090730_

Round 1
Reviewer 1 Report
Authors used a simple method for producing the porous carbons from marine algae by involving several steps of thermal treatment, and studied those carbon materials as ORR electrocatalysts. The reviewer suggests to address the following points:
Authors need to check the calculation for number of electrons transferred! The number of electrons transferred reported in this manuscript are more than 4 in some cases, however, a proper explanation is missing. The fittings for XP spectra in Figure 8 are not reliable due to the poor signal-to-noise ratio of the spectra. Therefore, the results in Table are also not reliable. The reference [31] cited in Table 1 doesn’t match the content.
Author Response
Dear Editors and Reviewers:
Thank you for your letter and for the comments concerning our manuscript entitled “Marine algae-derived porous carbons as robust electrocatalysts for ORR” (Manuscript ID: catalysts-582186). Those comments are all valuable and very helpful for revising and improving our paper, as well as the important guiding significance to our researches. We have studied comments carefully and have made correction which we hope meet with approval. Revised portion are marked in red in the paper. The main corrections in the paper and the responses to the reviewer’s comments are listed in the response letter point by point.
Best regards
Yours sincerely
Xianhua Liu
Response to reviews:
Referee: 1
Comments:
Authors used a simple method for producing the porous carbons from marine algae by involving several steps of thermal treatment, and studied those carbon materials as ORR electrocatalysts. The reviewer suggests to address the following points:
Authors need to check the calculation for number of electrons transferred! The number of electrons transferred reported in this manuscript are more than 4 in some cases, however, a proper explanation is missing.
Response:
Thank you for the comment. We agree with the reviewer that theoretically the number of electrons transferred to O2 is no more than 4. However, RDE measurement assumes certain conditions in the hydrodynamics. When working with thick layers of catalysts, such as porous carbon catalysts, the roughness of the surface may complicate these analyses. Furthermore, some oxides in the catalysts may capture electrons. These cases will make the experimental results deviate from the ideal results to a certain extent. This phenomena has also been reported in some other reports. For example, Chatterjee K, et al. Nitrogen-rich carbon nano-onions for oxygen reduction reaction [J]. Carbon, 2018, 130:645-651.
The fittings for XPS spectra in Figure 8 are not reliable due to the poor signal-to-noise ratio of the spectra. Therefore, the results in Table are also not reliable.
Response:
Thank you for the comment. Poor signal-to-noise ratio is indeed a problem when measuring XPS spectra, and this problem is aggravating when the element content is low. This problem has also been reported in some other reports.
1) Wu M, et al. N/S-Me (Fe, Co, Ni) doped hierarchical porous carbons for fuel cell oxygen reduction reaction with high catalytic activity and long-term stability[J]. Applied Energy, 2016, 175: 468-478.
2) Fan W, et al. Binary Fe, Cu-doped Bamboo-like Carbon Nanotubes as Efficient Catalyst for the Oxygen Reduction Reaction[J]. Nano Energy, 2017, 37:187-194.
The reference [31] cited in Table 1 doesn’t match the content.
Response:
Thank you for the comment. We have corrected the citation in our revised manuscript.
We have tried our best to improve the manuscript and made the corresponding corrections according to the comments. These corrections have been marked with red color in the revised manuscript. We appreciate for your warm work earnestly, and hope that the corrections will meet with approval. Once again, thank you very much for your comments and suggestions.
Best regards
Dr. Prof. Xianhua Liu
School of Environmental Scicence & Engineering
Tianjin University

Reviewer 2 Report
The manuscript entitled “Marine algae-derived porous carbons as robust electrocatalyst for ORR”
studies cobalt doped carbon prepared from algal as catalyst for oxygen reduction reactions in fuel cells. The work shows that natural algal biomass is an interesting and promising candidate for replacing precious metal catalysts with comparable or even better electrochemical performance. The results can be possibly a step towards a sustainable transformation for fuel cells.
In general the work is well described and presented in the manuscript, and the content is understandable also readers from related fields. A minor error is in Figure 3A, the labels on the y-axis are missing.
I do not feel sufficiently qualified on this topic to judge about the in-depth scientific soundness and technical correctness of the work. However, the work appears solid and I think it can be recommended for publication.
Author Response
Dear Editors and Reviewers:
Thank you for your letter and for the comments concerning our manuscript entitled “Marine algae-derived porous carbons as robust electrocatalysts for ORR” (Manuscript ID: catalysts-582186). Those comments are all valuable and very helpful for revising and improving our paper, as well as the important guiding significance to our researches. We have studied comments carefully and have made correction which we hope meet with approval. Revised portion are marked in red in the paper. The main corrections in the paper and the responses to the reviewer’s comments are listed in the response letter point by point.
Best regards
Yours sincerely
Xianhua Liu
Response to reviews:
Referee: 2
Recommendation: Recommended for publication.
Comments:
The manuscript entitled “Marine algae-derived porous carbons as robust electrocatalyst for ORR” studies cobalt doped carbon prepared from algal as catalyst for oxygen reduction reactions in fuel cells. The work shows that natural algal biomass is an interesting and promising candidate for replacing precious metal catalysts with comparable or even better electrochemical performance. The results can be possibly a step towards a sustainable transformation for fuel cells.
In general the work is well described and presented in the manuscript, and the content is understandable also readers from related fields.
A minor error is in Figure 3A, the labels on the y-axis are missing.
Response:
Thank you for the suggestion. The Y-axis is not labeled is because stacked cyclic voltammograms are used in order to show the reduction peaks of different catalysts more clearly and intuitively, which is helpful to compare the performance differences of catalysts. This presentation method has also been used in other reports. For example,
1)Liang Y, et al. Co3O4 nanocrystals on graphene as a synergistic catalyst for oxygen reduction reaction[J]. Nature Materials, 2011, 10(10):780-786.
2)Wu M, et al. N/S-Me (Fe, Co, Ni) doped hierarchical porous carbons for fuel cell oxygen reduction reaction with high catalytic activity and long-term stability[J]. Applied Energy, 2016, 175: 468-478.
3) Adam F. Henwood, et al. Palladium(0) NHC complexes: a new avenue to highly efficient phosphorescence. Chemical Science, 2015, 6, 3248-3261
We have tried our best to improve the manuscript and made the corresponding corrections according to the comments. These corrections have been marked with red color in the revised manuscript. We appreciate for your warm work earnestly, and hope that the corrections will meet with approval. Once again, thank you very much for your comments and suggestions.
Best regards
Dr. Prof. Xianhua Liu
School of Environmental Scicence & Engineering
Tianjin University
